# EQUIVARIANT NEURAL FIELDS FOR SYMMETRY PRESERVING CONTINUOUS PDE FORECASTING

**David M. Knigge***
VisLab
University of Amsterdam
d.m.knigge@uva.nl

**David R. Wessels***
AMLab
University of Amsterdam
d.r.wessels@uva.nl

**Riccardo Valperga**
VisLab
University of Amsterdam

**Samuele Papa**
POP-AART Lab
University of Amsterdam

**Efstratios Gavves**
QUVA Lab
University of Amsterdam

**Erik J. Bekkers**
AMLab
University of Amsterdam

## ABSTRACT

Recently, Neural Fields (NeFs) have emerged as a powerful modelling paradigm to represent discretely-sampled continuous signals. As such, novel work has explored the use of Conditional NeFs to model PDEs, by learning continuous flows in the latent space of the Conditional NeF. Although this approach benefits from favourable properties of neural fields such as grid-agnosticity and space-time-continuous dynamics modelling, it does not make use of important geometric information about the domain of the PDE being modelled – such as information on symmetries of the PDE – in favour of modelling flexibility. Instead, we propose a NeF parameterization that preserves geometric information in the latent space of the Conditional NeF: *Equivariant Neural Fields*. Using this representation, we construct a framework for space-time continuous PDE modelling that preserves known symmetries of the PDE. We experimentally validate our model and show it readily generalizes to arbitrary locations, as well as geometric transformations of the initial conditions - where other NeF-based PDE forecasting methods fail.

## 1 INTRODUCTION

Partial Differential Equations (PDEs) are a foundational tool in modelling and understanding spatio-temporal dynamics across diverse scientific domains. Classically, PDEs are solved using numerical methods like finite elements, finite volumes, or spectral methods. In recent years, Deep Learning (DL) methods have emerged as promising alternatives due to abundance of observed and simulated data as well as the accessibility to computational resources, with applications ranging from fluid simulations and weather modelling (Yin et al., 2022; Brandstetter et al., 2022a) to biology Moser et al. (2023). The systems modelled by PDEs often have underlying symmetries. For example, heat diffusion or fluid dynamics can be modeled with differential operators which are rotation equivariant, i.e. given a solution to the system of PDEs, its rotation is also a valid solution. In such scenarios it is sensible, and even desirable, to design neural networks that incorporate and preserve such symmetries by design to improve generalization and data-efficiency (Cohen & Welling, 2016; Weiler & Cesa, 2019; Bekkers, 2019).

Crucially, DL-based approaches rely on data sampled on a grid, without the inherent ability to generalize outside of it, which is restrictive in many scenarios (Prasthofer et al., 2022). To this end, Yin et al. (2022) propose to use Neural Fields (NeFs) for modelling and forecasting PDE dynamics. However, they fail to leverage the well-known equivariances present in the data.

We introduce a new class of $SE(n)$-equivariant conditional neural fields. We condition the NeF through a point cloud of latent feature vectors defined over the symmetry group of interest, disentangling *pose* and *appearance* in the latent space of functions. We subsequently propose a framework for solving PDEs by learning traversals of the latent space of this equivariant NeF with an equivariant Neural ODE, see Fig. 1 for an illustration.

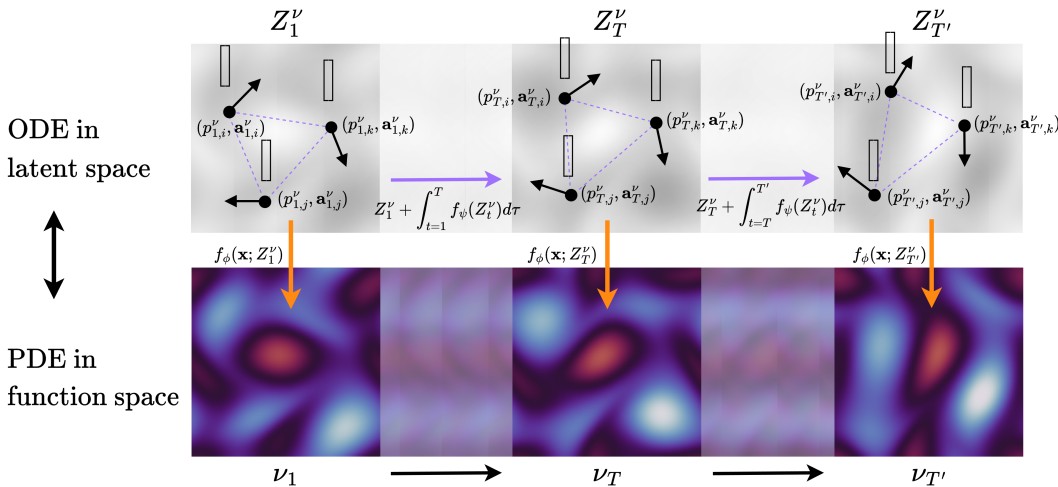

Figure 1: We propose to solve an equivariant PDE in function space by solving an equivariant ODE in latent space. Through our proposed *Equivariant Neural Field* $f_\theta$, a field $\nu_t$ can be represented by latent point clouds $Z_t^\nu = \{(p_{t,i}^\nu, \mathbf{a}_{t,i}^\nu)\}_{i=1}^T$ over symmetry group $SE(2)$. A PDE flow (bottom) corresponds to a trajectory of point clouds in latent space (top), generated by a learned neural ODE.

## 2 BACKGROUND AND RELATED WORK

We describe Deep Learning based approaches to data-driven dynamics modelling, and specifically highlight the recent use of neural fields (NeFs) in this setting. We highlight the concept of symmetry conservation in general DL model design and argue for the inclusion of such inductive biases in DL-based PDE surrogates, which we achieve by introducing equivariant conditional NeFs.

**DL approaches to dynamics modelling**  Most DL methods for solving PDEs attempt to directly replace solvers with mappings between finite-dimensional Euclidean spaces, i.e. through the use of CNNs (Guo et al., 2016; Ayed et al., 2020) or GNNs (Pfaff et al., 2020; Brandstetter et al., 2022b) often applied autoregressively to an observed (discretized) PDE state. Instead, the Neural Operator (NO) (Kovachki et al., 2021) paradigm attempts to learn infinite-dimensional operators, i.e. mappings between function spaces, with limited success. Fourier Neural Operator (FNO) (Li et al., 2020) extends this method by performing convolutions in the spectral domain. FNO obtains much improved performance, but due to its reliance on FFT is limited to data on regular grids.

**Inductive biases in DL and dynamics modelling**  The field of Geometric Deep Learning focuses on constraining/designing a model's space of learnable functions based on geometric principles to obtain improved performance and better generalization. Prominent examples from Computer Vision research include Group Equivariant Convolutional Networks Cohen & Welling (2016); Bekkers (2019) and Graph Neural Networks (GNNs) Kipf & Welling (2016), generalizations of CNNs that respect symmetries of the data - such as dilations and continuous rotations (Weiler & Cesa, 2019; Finzi et al., 2020; Knigge et al., 2022). In the context of dynamics modelling, equivariant architectures have been employed to incorporate various properties of physical systems in the modelling process, examples of such properties are the symplectic structure (Jin et al., 2020), discrete symmetries such reversing symmetries (Valperga et al., 2022) and energy conservation (Greydanus et al., 2019; Hernández et al., 2021).

**Neural Fields in Dynamics Modelling**  Neural fields (NeFs) are a class of coordinate-based neural networks, often trained to reconstruct discretely-sampled input signals in a continuous way. More specifically, a neural field $f_\theta : \mathbb{R}^n \to \mathbb{R}^d$ is a field –parameterized by a neural network with parameters $\theta$– that maps $n$-dimensional input coordinates $\mathbf{x} \in \mathbb{R}^n$ in the data domain, to $d$-dimensional signal values $f(\mathbf{x}) \in \mathbb{R}^d$. By associating a conditioning latent $\mathbf{z}^f \in \mathbb{R}^c$ to each signal $f$, a single conditional NeF $f_\theta : \mathbb{R}^n \times \mathbb{R}^c \to \mathbb{R}^d$ can learn to represent families $\mathcal{D}$ of continuous signals such that $\forall f \in \mathcal{D} : f(\mathbf{x}) \approx f_\theta(\mathbf{x}; \mathbf{z}^f)$. Dupont et al. (2022) showed the viability of using the latents $\mathbf{z}^f$ as representations for downstream tasks (e.g. classification, generation) proposing a framework for *learning on neural fields*. This framework inherits desirable properties of neural

fields, such as inherent support for sparsely and/or irregularly sampled data, and independence to signal resolution. Yin et al. (2022) propose to use conditional NeFs for PDE modelling by learning a continuous flow in the latent space of a conditional neural field. In particular, a set of latents $\{\mathbf{z}_i^\nu\}_{i=1}^T$ are obtained by fitting a conditional neural field to a given set of observations $\{\nu_i\}_{i=1}^T$ at timesteps $1, ..., T$; simultaneously, a neural ODE (Chen et al., 2018) is trained with those latents such that solutions correspond to the trajectories traced by the learned latents.

## 3 METHOD

**Mathematical preliminaries** Given a group $G$ with identity element $e \in G$, and a set $X$, the *group action* is a map $\mathcal{L}_g : G \times X \to X$ such that $\mathcal{L}_e(x) = x$ and $\mathcal{L}_{gh}(x) = \mathcal{L}_g(\mathcal{L}_h(x))$. In particular, we are interested in the Special Euclidean group $SE(n) = T_n \rtimes SO(n)$. $SE(n)$ is the roto-translation group consisting of elements $g = (\mathbf{t}, \mathbf{R})$ with group operation $g\,g' = (\mathbf{t}, \mathbf{R})(\mathbf{t}', \mathbf{R}') = (\mathbf{R}\mathbf{t}' + \mathbf{t}, \mathbf{R}\mathbf{R}')$; their action on function spaces is defined by $\mathcal{L}_g f(\mathbf{x}) = f(g^{-1}\mathbf{x}) = f(\mathbf{R}^{-1}(\mathbf{x} - \mathbf{t}))$. Group elements of $SE(n)$ are identified by a translation $\mathbf{t} \in T_n \equiv \mathbb{R}^n$ and rotations $\mathbf{R} \in SO(n)$. Laws of physics do not depend on the choice of coordinate system, this implies that many PDEs are defined by $SE(n)$-equivariant differential operators $\mathcal{N}$, i.e., such that $\mathcal{L}_g \mathcal{N}[f] = \mathcal{N}[\mathcal{L}_g f]$. The same holds for ordinary differential equations defined by equivariant vector fields, if $\frac{d\mathbf{z}}{d\tau} = F(\mathbf{z})$ is such that $F(l_g \mathbf{z}) = l_g F(\mathbf{z})$, where we denote the group action of $SE(n)$ on the state space with $l_g$.

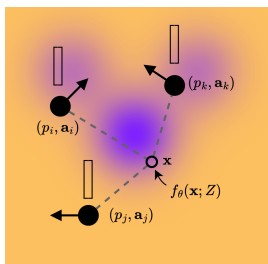

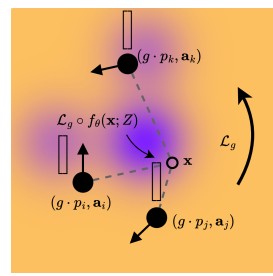

Figure 2: The proposed Equivariant Neural Field (top) under a roto-translation $g$ (bottom).

**PDE Modelling with Equivariant Neural Fields** We consider flows of fields, denoted with $\nu : \mathbb{R}^d \times [\![T]\!] \to \mathbb{R}^c$, in which $[\![T]\!] := 1, 2, \ldots, T$ denotes the set of time points on which the flow is sampled. A snapshot of the field at time index $t$ is denoted with $\nu_t$. We assume the flow is governed by a PDE which we aim to learn and represent as a neural ODE in latent space. To this end, we build upon the work of Yin et al. (2022), and consider the following optimization problem

$$\min_{\theta, \psi, \mathbf{z}_\tau} \quad \mathbb{E}_{\nu \in D, \mathbf{x} \in X, t \in [\![T]\!]} \|\nu_t(\mathbf{x}) - f_\theta(\mathbf{x}; \mathbf{z}_t^\nu)\|_2^2, \quad \text{where} \quad \mathbf{z}_t^\nu = \mathbf{z}_0^\nu + \int_0^t F_\psi(\mathbf{z}_\tau^\nu) d\tau, \quad (1)$$

where $F_\psi$ is a neural network that parametrizes the neural ODE, that is, it maps latent $\mathbf{z}_\tau^\nu \in \mathbb{R}^c$ to its temporal derivative: $\frac{d\mathbf{z}_\tau^\nu}{d\tau} = F_\psi(\mathbf{z}_\tau^\nu)$. Here $\mathbf{z}_t^\nu$ denotes the latent associated with $\nu_t$ and is inferred during training. Note that compared to Yin et al. (2022) where two independent objectives are minimised simultaneously (one for $\mathbf{z}_t^\nu$ and $\theta$ and one for $\psi$), we train our model with a *single* objective. Intuitively, this is to force the latents to not only reconstruct observations, but also to be arranged in such a way that they can be fitted by a continuous flow.

Our objective is to leverage the inductive bias of equivariance to preserve $SE(n)$ symmetries that a PDE may possess in our neural surrogate. This requires the following necessary conditions:

1. The latent space $Z$ is equiped with a well-defined group action $l_g$.

2. The relation between field and latent is equivariant, i.e., $\forall_{g \in SE(n)} : \mathcal{L}_g \nu_t \Leftrightarrow l_g Z_t^\nu$.

3. The neural ODE is equivariant, i.e., $\forall_{g \in SE(n)} : F_\psi(l_g \mathbf{z}_t^\nu) = l_g F_\psi(\mathbf{z}_t^\nu)$.

The next section describes how we solve the necessary conditions (1.) and (2.) by defining the latent space to a space of attributed point clouds in $SE(n)$ and introducing a *new class of $SE(n)$-equivariant conditional neural fields*. Condition (3.) will be satisfied by parametrizing the neural ODE $F_\psi$ with an equivariant graph neural network (Bekkers et al., 2023).

**Equivariant Conditional Neural Fields**   We define NeFs that are conditioned on attributed point clouds over $SE(n)$. That is, as a conditioning variable we consider a set $Z := \{(p_i, \mathbf{a}_i)\}_{i=1}^n$ of $n$ tuples that consist of a *pose* $p_i \in SE(n)$ and *appearance* $\mathbf{a}_i \in \mathbb{R}^c$. In clearer terms, each field $f$ will be associated with a latent set $Z$ through the NeF such that $f(\mathbf{x}) \approx f_\theta(\mathbf{x}; Z^f)$. The latent space has a well defined action $l_g$ given by $l_g Z = \{(g \cdot p_i, \mathbf{a}_i)\}_i^n$, with $g \cdot p_i$ being simply the group operation of $SE(n)$.

We propose a *new class of equivariant neural fields* (Fig. 2) that posses the following steerability property

$$\boxed{\forall g \in SE(n): \quad f_\theta(g^{-1}\mathbf{x}; Z) = f_\theta(\mathbf{x}; l_g\, Z)} \tag{2}$$

Through this property, a roto-translation of the field can be obtained by a roto-translation of the latent point cloud. We define the equivariant neural field through cross-attention between relative coordinates $p_i^{-1}\mathbf{x} = \mathbf{R}_i^T(\mathbf{x} - \mathbf{x}_i)$ and the appearance vectors $\mathbf{a}_i$ via

$$f_\theta(\mathbf{x}; Z) := \sum_{i=0}^n v(\mathbf{a}_i) \cdot \text{softmax}\left(\frac{q(p_i^{-1}\mathbf{x})^T k(\mathbf{a}_i)}{\sqrt{d_k}}\right), \tag{3}$$

in which $q$ is a learnable relative coordinate embedding function (we use SIREN from Sitzmann et al. (2020)), and $k, v$ are learnable linear transformations. The softmax is over the latent set, and $d_k$ represents the dimensionality of the embeddings. Note that $q$ can be any type of function that is not necessarily equivariant since the quantity $p_i^{-1}\mathbf{x}$ is bi-invariant under the group action via

$$\forall g \in SE(n): \ (p_i, \mathbf{x}) \mapsto (g\, p_i, g\, \mathbf{x}) \ \Leftrightarrow \ p_i^{-1}\mathbf{x} \mapsto (g\, p_i)^{-1} g\, \mathbf{x} = p_i^{-1} g^{-1} g\, \mathbf{x} = p_i^{-1}\mathbf{x}. \tag{4}$$

This also proves the steerability property (2) since by (4) the tuple $g^{-1}\mathbf{x}, p_i$ is equivalent to $\mathbf{x}, g\, p_i$.

## 4   Experiments

We validate our model on two different PDEs. First, to experimentally validate the equivariance properties of our framework, we train our models on 2D diffusion $\frac{dc}{dt} = D\nabla^2 c$, where $c$ is a scalar field, and $D$ is the diffusivity. Specifically, we create a benchmark where for initial conditions we insert a pulse at a random location in the grid $x = (x_1, x_2) \in \mathbb{R}^2$ s.t. $-1 < x_1 < 1, 0 < x_2 < 1$ for the training data and $-1 < x_1 < 1, -1 < x_2 < 0$ for the test data (intuitively, train and test sets contain pulses under different group actions respectively). We also evaluate on 2D Navier Stokes (Stokes et al., 1851) corresponding to an incompressible fluid with dynamics $\frac{dv}{dt} = -u\nabla v + v\Delta\mu + f, v = \nabla \times u, \nabla u = 0$, where $u$ is the velocity field, $v$ the vorticity, and $\mu$ the viscosity. For $f$ we choose a symmetric forcing term. For dataset details see Appx. A.1, for experimental details see Appx. A.2.

The models are trained and evaluated with the MSE on intervals of $T = 20$ steps split into $t_{\text{IN}} \in \{1, ..., 10\}$ and $t_{\text{OUT}} \in \{10, ..., 20\}$. Only $t_{\text{IN}}$ is used for supervision with the loss described in Eq. 1. Following Yin et al. (2022), we additionally provide results for randomly subsampled initial states $\nu_1$, where we show very little degradation even when $95\%$ of values are removed from the observed initial state of a trajectory. A visualization of a predicted train and test trajectory for both datasets may be found in Appx. A.3.

On the diffusion dataset (Tab. 1), our model obtains good performance on both train and test sets. In this experiment, DINo fails to generalize to the test set due to its lack of symmetry-preservation. On Navier-Stokes with symmetric forcing (Tab. 2), we similarly obtain good results on both train and test sets, although performance degrades for timesteps $t_{\text{out}}$. We conclude that symmetry-preservation improves space-time continuous PDE solving.

## 5   Conclusion

In this paper we proposed a model which utilises pointclouds as conditioning terms for conditional neural fields. Using an equivariant neural ode parameterised with a $SE(n)$-equivariant GNN we showed that utilising these conditional local latent point clouds within PDE-modeling outperforms the baseline DINo in two different experiments.

Table 1: MSE ↓ on roto-translated 2D Diffusion.

| 2D Diffusion | | | | |
| --- | --- | --- | --- | --- |
| | $t_{\text{IN}}$ TRAIN | $t_{\text{OUT}}$ TRAIN | $t_{\text{IN}}$ TEST | $t_{\text{OUT}}$ TEST |
| DINo | 5.922e-04 | **2.400e-04** | 3.849e-03 | 5.115e-03 |
| Ours | **2.547e-05** | 9.808e-04 | **6.424e-05** | **9.221e-04** |
| 50% OBSERVED | | | | |
| DINo | 4.132e-04 | **2.920e-04** | 3.977e-03 | 5.144e-03 |
| Ours | **7.000e-05** | 2.004e-03 | **9.721e-05** | **2.066e-03** |
| 5% OBSERVED | | | | |
| DINo | 7.568e-04 | **7.160e-04** | 5.786e-03 | 6.861e-03 |
| Ours | **9.000e-05** | 2.000e-03 | **1.034e-04** | **2.159e-03** |

Table 2: MSE ↓ on 2D Navier-Stokes.

| 2D Navier Stokes | | | | |
| --- | --- | --- | --- | --- |
| | $t_{\text{IN}}$ TRAIN | $t_{\text{OUT}}$ TRAIN | $t_{\text{IN}}$ TEST | $t_{\text{OUT}}$ TEST |
| DINo | 2.585e-01 | 5.477e-01 | 5.381e-01 | 8.988e-01 |
| Ours | **2.009-e02** | **1.429e-01** | **4.030e-02** | **2.029e-01** |
| 50% OBSERVED | | | | |
| DINo | 4.463e-01 | 7.632e-01 | 4.807e-01 | 8.719e-01 |
| Ours | **4.557e-02** | **1.812e-01** | **4.454e-02** | **2.400e-01** |
| 5% OBSERVED | | | | |
| DINo | 4.717e-01 | 7.792e-01 | 4.658e-01 | 8.642e-01 |
| Ours | **5.541e-02** | **2.086e-01** | **4.475e-02** | **2.369e-01** |

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

# A APPENDIX

## A.1 DATASET DETAILS

**Diffusion 2D**  We generate rollouts of 2048 initial conditions for use as training data using an RK4 solver, and an additional 128 as test data. We sample initial conditions by selecting a random point $\mathbf{x}$ in the domain $[-1, 1]^2$ (for training data $-1 < x_1 < 1, 0 < x_2 < 1$, for test data $-1 < x_1 < 1, -1 < x_2 < 0$) and inserting a random value sampled from $U[0.75, 1.25]$. The full spatial grid is sampled at 64x64 resolution. We discard the initial five timesteps.

**Navier-Stokes 2D**  We generate rollouts of 8192 initial conditions for use as training data using an RK4 solver, and an additional 128 as test data. The spatial domain is set to $\Omega = [-1, 1]^2$, viscosity is $1e^{-3}$ and forcing term $f$ is set as:

$$\forall x \in \Omega, f(x_1, x_2) = 0.3\big(\cos(4\pi x_1) + \cos(4\pi x_2)\big) \tag{5}$$

The full spatial grid is sampled at 64x64 resolution.

## A.2 EXPERIMENTAL DETAILS

**Optimization**  For all experiments, we use the Adam optimization Kingma & Ba (2014), with learning rates $1e^{-4}$ for the neural ODE $F_\psi$ and Equivariant Neural Field $f_\theta$ and $1e^{-3}$ for the latents $\mathbf{z}$. All models are trained for 300 epochs. During testing and inference, we fit a newly initialized latent to the observed initial state for 300 epochs.

**Details on latents**  Latents are initialized as identity vectors, i.e. for all latents $i$ $\mathbf{a}_i = \mathbf{1}^c$ with $c$ the dimensionality of the latent. The poses $p_i$ are initialized on an equidistantly spaced grid over the domain and unit circle. We use 4 latents with $c = 32$ for all experiments we reported. Both latents and poses are optimized for a state using backpropagation.

**Details on equivariant neural field**  The $q, k, v$ transforms are all linear layers that map to a hidden dimensionality of 128. We apply 2 attention heads. The equivariant cross attention operation is followed by two fully connected layers of dimensionality 128, the final one mapping to the output dimensionality (which is 1 in all of our experiments).

## A.3 SAMPLE TRAJECTORIES

We show some randomly selected test and train trajectories generated by our model. We fit latents $Z$ to the initial state of the selected trajectory $\nu$, and unroll the neural ODE for 20 timesteps. In each of the figures, the top row corresponds to the ground truth, the second row corresponds to the predicted trajectory, the third row shows the position and orientation of the latent, and the final row shows the absolute difference between ground truth and reconstruction.

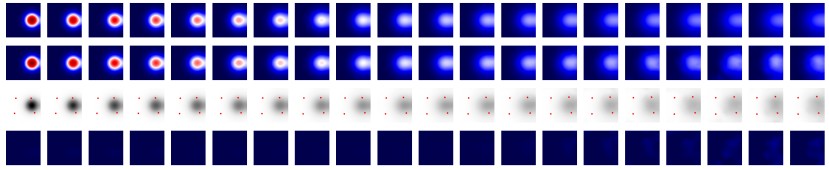

Figure 3: A predicted solution on the training set for the diffusion equation.

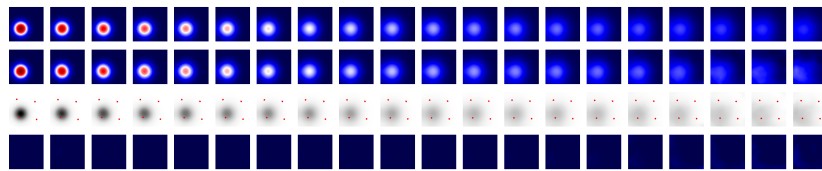

Figure 4: A predicted solution on the test set for the diffusion equation.

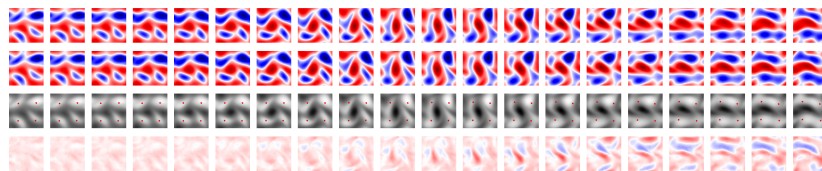

Figure 5: A predicted solution on the training set for the Navier-Stokes equation.

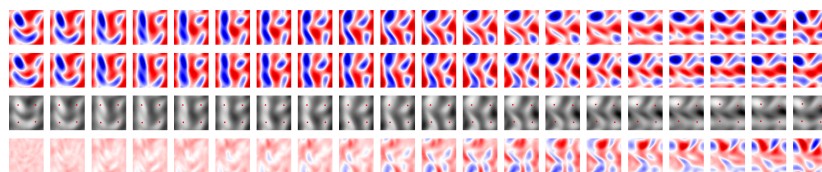

Figure 6: A predicted solution on the test set for the Navier-Stokes equation.

