# OpenReview forum: "Equivariant Neural Fields For Symmetry Preserving Continous PDE Forecasting"
_ICLR.cc/2024/Workshop/AI4DiffEqtnsInSci — AI4DiffEqtnsInSci @ ICLR 2024 Poster_

### Official Review · Reviewer_b57K · 2024-02-25
**Introduces SE(n) equivariant model to augment CNeFs with geometric properties such as roto-translation to achieve better generalization.**

**Rating:** 6
**Confidence:** 4

**Review:**

Existing neural fields architecture for PDE solving lack components that preserve symmetries in data. And this work introduces an SE(n) equivariant approach to augment CNeFs with geometric properties such as roto-translation.

**Strengths**:
- generalization to geometric transformations of ICs
- learning equivariant latent evolution
- Roto-Translation in the latent space

**Weaknesses**:
- Limited set of experiments.
- No separate validation set. Hence the model might have been tuned on the test set.

**Questions**:
- How was the 50% and 5% observed points selected?
- How to decide which PDEs possess SE(n) symmetries?
- How to choose the optimal number of latents?

**Suggestions for improvement**:
- The importance of steerability on high dimensional PDEs can be emphasized.

---

### Official Review · Reviewer_hBmb · 2024-02-27
**Equivariant latent fields**

**Rating:** 4
**Confidence:** 5

**Review:**

The authors propose using an equivariant point cloud representation for latent variables in a neural PDE formulation. The idea is tested on 2D diffusion and 2D vorticity equations with rotationally symmetric forcing.

The numerical results show improvement based on the baseline, but the baseline DINo is never cited or defined in text. The authors do not provide details on the ground truth solver used or how the initial conditions for vorticity equation was generated. They do not provide value for diffusivity D and do not state what the time steps used in evaluation are, making it impossible to judge the results. Finally, there are 2 typos in the statement of 2d NS equation: it is \mu \Delta v and not \nabla u = 0 but \nabla \cdot u = 0.

In addition to lack of detail, the experiments are very contrived: what happens if forcing f is not rotationally symmetric, as it is most practical applications?

---

### Meta-Review · Area_Chair_HYbL · 2024-02-28

**Recommendation:** Accept (Poster)

**Metareview:**

Dear Authors,

Thank you for submitting the draft.

Both reviewers agree that the presented work presents some interesting strengths. However, both reviewers do also raise some major points of concern, especially regarding the clarity of the presentation. It is expected that authors will be addressing comments by the reviewers in the final draft.

regards

AC

---

### Decision · Program_Chairs · 2024-02-29

Accept (Poster)